# Morphokinetic Profiling Suggests That Rapid First Cleavage Division Accurately Predicts the Chances of Blastulation in Pig In Vitro Produced Embryos

**DOI:** 10.3390/ani14050783

**Published:** 2024-03-02

**Authors:** Lucy M. Hillyear, Louisa J. Zak, Tom Beckitt, Darren K. Griffin, Simon C. Harvey, Katie E. Harvey

**Affiliations:** 1School of Psychology and Life Sciences, Canterbury Christ Church University, Canterbury CT1 1QU, UK; l.vining115@canterbury.ac.uk; 2Topigs Norsvin Research Center, Meerendonkweg 25, 5216 TZ Den Bosch, The Netherlands; louisa.zak@topigsnorsvin.com; 3Genea Biomedx, Sydney, NSW 2000, Australia; tom.beckitt@geneabiomedx.com; 4School of Biosciences, University of Kent, Canterbury CT2 7NJ, UK; 5Faculty of Engineering and Science, University of Greenwich, Medway ME4 4TB, UK; s.c.harvey@greenwich.ac.uk; 6School of Life, Health and Chemical Sciences, The Open University, Milton Keynes MK7 6AA, UK; katie.harvey@open.ac.uk

**Keywords:** blastocyst, morphokinetics, pig, predictive parameters, time-lapse

## Abstract

**Simple Summary:**

The study of the earliest stages of pig embryo development has several potential uses: feeding a growing population, medical applications that require the use of genetically modified organisms, rare breed conservation, and the use of pig embryos as a research model for progressing IVF technology in both humans and other (e.g., endangered) species. Despite this, barriers to the widespread adoption of pig embryo production include fat-rich cells that are difficult to see down a microscope, low success rates, and slow adoption of new technologies such as artificial intelligence. Here, we fully characterise pig early embryo development in an undisturbed system, using videos derived from an incubator equipped with in-built cameras. We find that a rapid first cell division, from one to two cells, correlates with ongoing embryo success and that this early-stage measure alone can be an accurate predictor of future development. Conversely, we find that re-fusion of cells following division is associated with poor rates of embryo development. Such predictors may allow for the early commercial selection of high-quality embryos and upscaling of production so that the true potential of pig embryology can be realised.

**Abstract:**

The study of pig preimplantation embryo development has several potential uses: from agriculture to the production of medically relevant genetically modified organisms and from rare breed conservation to acting as a physiologically relevant model for progressing human and other (e.g., endangered) species’ in vitro fertilisation technology. Despite this, barriers to the widespread adoption of pig embryo in vitro production include lipid-laden cells that are hard to visualise, slow adoption of contemporary technologies such as the use of time-lapse incubators or artificial intelligence, poor blastulation and high polyspermy rates. Here, we employ a commercially available time-lapse incubator to provide a comprehensive overview of the morphokinetics of pig preimplantation development for the first time. We tested the hypotheses that (a) there are differences in developmental timings between blastulating and non-blastulating embryos and (b) embryo developmental morphokinetic features can be used to predict the likelihood of blastulation. The abattoir-derived oocytes fertilised by commercial extended semen produced presumptive zygotes were split into two groups: cavitating/blastulating 144 h post gamete co-incubation and those that were not. The blastulating group reached the 2-cell and morula stages significantly earlier, and the time taken to reach the 2-cell stage was identified to be a predictive marker for blastocyst formation. Reverse cleavage was also associated with poor blastulation. These data demonstrate the potential of morphokinetic analysis in automating and upscaling pig in vitro production through effective embryo selection.

## 1. Introduction

Assisted reproduction technology (ART) in domestic animals, principally in vitro production (IVP), can be used to accelerate rates of genetic progress (reducing generational intervals and enhancing selection intensity) and move genetics across international borders in a cost-effective, environmentally friendly, disease-free manner. In some livestock species, it can promote animal health, welfare, and disease control during stock changeover [1]. Further benefits include the generation of resources for bio-banking (maintaining biodiversity by preserving rare breeds, lines, or endangered species), the use of embryos as models for human IVF research, and gaining a greater insight into the fundamental biological processes involved in mammalian preimplantation embryo development [2]. Each species in which IVP has been attempted has had its own peculiarities identified, and, in the case of pigs, progress has lagged behind that of cattle and humans as a result of lipid-laden cells that are hard to visualize and cryopreserve. Lower blastulation rates, high levels of polyspermy, and a much longer reproductive tract (among other challenges) are also significant barriers to the adoption of this technology in pigs [2], even though it has great potential to facilitate faster and more precise pig breeding programmes. The selection of high-quality embryos from genetically superior animals may, in fact, accelerate the production of animals with desirable traits such as improved meat quality, disease resistance, and/or reproductive efficiency [2]. Increased interest in producing genetically modified pigs for purposes such as xenotransplantation [3] and studying human disease to facilitate the development of novel therapies [4,5] also requires an active pig IVP programme.

Successful embryo transfer in any species relies on the generation and selection of embryos with the highest likelihood of producing healthy offspring, and good morphological criteria are often the first port of call in determining the embryos most likely to proceed to live births [6,7,8,9]. Selection by morphology, however, traditionally requires the removal of embryos from the incubator at pre-determined time intervals during embryo culture [10,11]. Although this can provide an important snapshot of embryo morphokinetics, only limited data can be collected [12]. Moreover, the removal of embryos from the optimal environment of the incubator has the potential to disrupt their development due to fluctuations in oxygen tension, temperature, light, and pH [13,14]. Therefore, uninterrupted culture through time-lapse technology has many theoretical advantages over a traditional, well-managed embryo culture incubator with embryo evaluation. However, the total consensus and/or evidence as to the magnitude of the potential benefits is yet to be reached.

Morphokinetic analysis through the use of time-lapse technology is now widely used in human IVF, with several different systems available commercially [12]. This facilitates continuous, undisturbed embryo observation and both prospective and retrospective studies of early preimplantation development. Such work has demonstrated that morphokinetic data can be used to predict blastocyst formation [15,16,17], implantation [9,18], pregnancy [19,20] and live birth rates [21]. As a result, increasingly complex embryo selection algorithms, often involving artificial intelligence (AI), that can be used in embryo selection have been generated [22,23,24,25]. Morphokinetic analysis also allows for the detection of abnormal embryonic developmental events, some of which are difficult or impossible to discover using traditional incubation and occasional visual inspection. Examples of aberrant developmental events include high levels of fragmentation, the appearance of uneven blastomeres, cell extrusion, chaotic cleavage (i.e., extensive fragmentation at the first cleavage resulting in no distinguishable blastomeres), direct cleavage (the cleavage of one blastomere to more than two blastomeres), and reverse cleavage (fusion of two blastomeres). These developmental abnormalities have all been associated with the incidence of chromosomal aberrations [26,27,28,29] and with decreased rates of blastocyst formation in humans [30].

However, the relevance of such morphokinetic abnormalities in predicting embryo developmental failure in other species remains unclear as few morphokinetic studies have been undertaken in non-human animals, as reviewed in Mandawala et al., 2016. The field has remained largely unchanged since the publication of this paper, although Wiesak et al. (2021) provided an update [14,31,32]. Therefore, the multi-faceted potential of pig embryology and the ready availability of time-lapse devices provide an opportunity to redress this imbalance. Indeed, automated embryo selection could allow IVP to reach its full potential, upscaling production to the volumes required to make it commercially effective.

With the above in mind, this study has three objectives, all of which employ a commercially available time-lapse incubator designed for human studies. The first was to provide a comprehensive overview of the morphokinetics of pig preimplantation development. The second was to test the hypothesis that there are differences in the developmental timings between blastulating and non-blastulating IVP pig embryos. The third was to test the hypothesis that embryo developmental morphokinetic features can be used to predict the likelihood of blastulation.

## 2. Materials and Methods

### 2.1. Chemicals and Reagents

All chemicals and reagents were purchased from Sigma-Aldrich (Gillingham, Dorset, UK) unless otherwise stated. All stages of IVP were performed in a Geri^®^ incubator (Genea Biomedx, Sydney, NSW, Australia) with standard gas (6% CO_2_, 5% O_2_) (BOC, Woking, Surrey, UK) in a non-humidified environment at 38.5 °C.

### 2.2. Oocyte Collection and In-Vitro Maturation (IVM)

Six rounds of IVP were performed in this study. In each case, predominately prepubertal gilt ovaries were collected from an abattoir and placed inside a sealed plastic bag suspended in water (35 °C) inside a thermos flask before transportation to the laboratory within six hours. Before aspiration, ovaries were washed three times in phosphate-buffered saline and placed into a 28 °C water bath. Cumulus-oocyte complexes (COCs) were collected from follicles ranging from 3 to 8 mm using a syringe (Norm-Ject^®^, Henke Sass Wolf, Tuttingen, Germany) equipped with an 18 G needle [33]. The aspirated follicular contents were allowed to sediment in Falcon tubes before washing three times in Porcine X Medium (PXM) [34] and pre-warmed to 38 °C. COCs with three layers of cumulus cells and an even cytoplasm were selected and washed three times in Porcine Oocyte Medium (POM) [34]. For IVM, COCs were cultured in POM supplemented with FSH (0.5 IU/mL), LH (0.5 IU/mL), dbCAMP (0.1 mM) and FLI [35] for 20 h. Subsequently, COCs were cultured in POM, only supplemented with FLI for a further 24 h.

### 2.3. In-Vitro Production (IVP) of Pig Embryos

For each of the six rounds of IVP, an extended boar semen sample (JSR Genetics Ltd., Driffield, East Yorkshire, UK) was prepared for use in IVF using BoviPure™ density gradient centrifugation (Nidacon, Mölndal, Sweden) as per the manufacturer’s instructions for ‘Double Layer Fresh Semen’. After IVM, all oocytes were washed twice in Porcine Gamete Medium (PGM) [34] and co-incubated with 100,000 sperm/mL for two hours. Oocytes were then moved to another fresh well of PGM for a further two hours to reduce the risk of polyspermy. Following this, presumptive zygotes were denuded by repeated aspiration using a P200 pipette.

### 2.4. In-Vitro Embryo Culture in a Time-Lapse Incubator

Denuded presumptive zygotes were placed into Geri^®^ (Genea Biomedx, Sydney, NSW, Australia) dishes that allow the co-culture of up to 16 embryos in microwells in a single dish. These were prepared with an 80 µL droplet of culture media overlaid with 4 mL of mineral oil. The embryos were then placed in a Geri^®^ incubator for 144 h post-insemination (hpi). Intra-user variability of embryo annotation was performed to assess the variability of a single user performing the analysis. The analysis of developmental events of embryos reaching the 2-cell, 3-cell, 4-cell, 5+ cell, compaction, morula, cavitation/early blastocyst, expanding blastocyst and hatching blastocyst stages were included. Two-way intraclass correlation coefficient (ICC) values were interpreted as follows: <0.5 poor agreement, 0.5–0.75 moderate agreement, 0.75–0.9 good agreement, 0.9+ excellent agreement.

### 2.5. Statistical Analysis

Data were analysed using RStudio (version 4.4.1). All data were assessed as non-parametric using the Shapiro–Wilk Normality test. A Mann–Whitney U test was used to determine statistical differences between developmental timings of blastulating and non-blastulating groups. An odds ratio chi-squared test was used to assess abnormal observations in both groups. Additionally, binomial general linear regression models were calculated for each individual developmental event, with approximately 50% of the data points randomly selected to produce training data sets. ROC curves were calculated using these models by applying the prediction model to a test training set, and AUC scores were then calculated.

## 3. Results

### 3.1. Assessment of the Intra-User Variability of Embryo Annotation

Validation of embryo staging by a single user was performed on a subset of 125 videos from two repeats of IVP containing both arresting and blastulating embryos. Of these, 70 embryos did not show any cell division and were therefore excluded; 55 embryos, 24 of which developed to the blastocyst stage, and 31 that did not, were compared. A listwise deletion of missing observations was performed to calculate a two-way ICC. As such, four more embryos were excluded; 51 embryos were therefore included for this ICC. The mean intra-observer agreement was 0.902, which can be interpreted as excellent agreement between the first (A) and second (B) analyses (Table 1). The lowest agreement was seen for hatching blastocysts at 0.438 (poor), followed by compaction at 0.772 (good). All other observations had ICC scores > 0.9, indicating excellent agreement.

### 3.2. Developmental Kinetics of Blastulating and Non-Blastulating Pig Embryos

Time-lapse data was collected from a total of 269 presumptive zygotes comprising 111 cleaved embryos (41.3%), and 47 embryos out of those that cleaved reached the blastocyst stage (42.3%). The average blastocyst rate per oocyte obtained was 17.5%. Representative examples of the key developmental events observed can be seen in Figure 1.

The boxplot in Figure 2A shows the number of hours post-insemination (hpi) taken to reach each developmental stage, while Figure 2B shows the progression of these individual embryos throughout development. Here, three patterns of blastocyst development are evident. Group (1): Highly synchronous cleavage from 2- to 4 cells, with a slowing of development between 4- to 5+ cells. Group (2): A long lag phase between 2- to 3-cells followed by rapid cell division. Group (3): Rapid initial divisions followed by a long lag phase between 5+ cells and the morula stage.

Addressing our first hypothesis, we observed significant differences in the time taken to reach the 2-cell stage in blastulating vs. non-blastulating embryos. That is, the time for the former was 36.34 ± 1.13 hpi (mean hours ± standard error) and 42.01 ± 3.38 hpi for the latter. The time taken to reach the morula stage was 97.03 ± 1.34 h for blastulating embryos and 109.99 ± 3.29 hpi for non-blastulating embryos (Figure 2A). Data were also collected for the stage at which embryo arrest occurred. Of the 222 non-blastulating embryos, 172 (77%) either arrested prior to the first cleavage or reverse cleaved to 1 cell, 11 (5%) arrested at the 2-cell stage, 10 (5%) arrested at 3-cells, 5 (2%) arrested at 4-cells, 12 (5%) arrested at 5–8 cells and 18 (5%) arrested at the morula stage.

### 3.3. Embryo Developmental Time as a Predictor of Blastocyst Formation

To further address the hypothesis that developmental timings are associated with embryo outcomes, binomial general linear regression models were calculated for each developmental event. All embryos that had an annotation for the specific developmental event were included. In training linear regressions, the 2-cell stage (Z = 2.53, *p* = 0.01) and the 3-cell stage (Z = 2.01, *p* = 0.04) showed that the time taken to reach the developmental stage was a significant predictor in the model. There was no significant relationship between the time taken to reach the 4-cell (Z = 1.64, *p* = 0.10), 5+ cell (Z = 1.34, *p* = 0.18) and morula (Z = 0.56, *p* = 0.58) stages and blastulation. ROC curves and respective AUC scores were calculated for each developmental event, with AUC scores being interpreted as: 0.5 = no discrimination, 0.7–0.8 = acceptable discrimination, 0.8–0.9 = excellent discrimination and >0.9 = outstanding discrimination. Excellent discrimination was found for the 2-cell and morula stages, and acceptable discrimination was found for the 4-cell stage. In binomial regression models, only the time taken to the 2-cell stage was shown to be a significant predictor of blastocyst formation with excellent discrimination (AUC > 0.8–0.9) (Figure 3).

In summary, only the 2-cell stage (*n* = 76) was a significant predictor for blastocyst outcomes in linear regression and showed excellent discrimination in ROC/AUC analysis. In an odds ratio chi-squared test, the embryos reaching the 2-cell stage before 30 hpi had the highest significant odds of developing into a blastocyst (OR = 10, *p* = 0.00004) (Table 2).

### 3.4. Abnormal Embryo Development as a Predictor of Blastocyst Formation

To extend this hypothesis to morphokinetic features other than the timing of cell division, odds ratio chi-squared tests were used to investigate the relationship between abnormal observations annotated during embryo development and blastocyst formation success. Odds ratios for embryos displaying any degree of fragmentation found no significant difference between blastulating and non-blastulating embryos (OR = 0.70, *p* = 0.41) (*n* = 269). However, a significantly higher number of blastocysts with both fragmentation up to 10% (OR = 0.10, *p* = <0.0005) and fragmentation between 11 and 25% (OR = 0.31, *p* = 0.003) were found when compared to non-blastulating embryos. Chaotic cleavage was not observed in any embryos that reached the blastocyst stage but occurred relatively commonly in our experiments, affecting 35% of all embryos and accounting for 43% of non-blastulating embryos (*n* = 95, OR = 72.3, *p* = 0.003). All embryos that displayed no cleavage or chaotic cleavage (lethal to embryo development) were excluded from further analyses. In the remaining embryos (*n* = 111), reverse cleavage was significantly more common in non-blastulating embryos than in those that reached the blastocyst stage (OR = 4.36, *p* = 0.03). There was no significant difference in the number of embryos displaying uneven blastomere size (OR = 1.32, *p* = 0.50), direct cleavage (3-cell) (OR = 2.31, *p* = 0.22), and 118 direct cleavage (≥4 cell) (OR = 0.48, *p* = 0.13) whether embryos reached the blastocyst stage or not. In embryos reaching the morula stage (*n* = 59), there was no significant difference in the number of embryos with cell extrusion between blastulating and non-blastulating embryos (OR = 0.96, *p* = 0.95). In the blastulating group (*n* = 47), 15 embryos (32%) showed blastocyst pulsing (seven pulsing only once, eight pulsing twice) within 144 hpi. Additionally, eight embryos (9%) showed blastocyst collapse, with four of these remaining collapsed during the time of recording.

## 4. Discussion

To the best of our knowledge, this is the most comprehensive assessment of the morphokinetics of the earliest stages of pig development to date. The use of a Geri^®^ time-lapse incubator system in place of conventional incubation allowed for undisturbed embryo development. The detailed observation and accurate assessment of developmental timings hitherto not possible were also achieved, with images being taken every five minutes.

Our observation that the time taken to reach both the 2-cell and the morula stages was significantly shorter in blastulating embryos (Figure 2A) concurs with other studies in a number of species that report that early first cleavage is associated with improved developmental potential. These include mice [36,37,38], humans [39,40], cattle [41], and pigs [11,31,42,43,44]. Further to this, IVP pig embryos that reach the early morula stage prior to 102 hpi have been found to be more likely to form a blastocyst [32]. In humans (and other species), the time to morula formation has been demonstrated to be able to be used as a predictor of pregnancy [45].

Here, we established that many blastocysts followed the expected developmental trajectory, i.e., highly synchronous cleavage between 2- and 4-cells and a lag phase from 4 to 5+ cells (Group 1, Figure 2B). This compares to studies in humans demonstrating that synchrony between the 3- and 4-cell stages is a potential predictor of blastocyst formation [17], implantation success [9] and normal chromosome complement [46]. The two other distinct developmental patterns that were observable (Figure 2B) were Group 2: Those that had a long lag phase between 2- and 3-cells followed by quick cell division, and Group 3: Those that displayed fast initial cell divisions, followed by a long lag phase between 5+ cells and the morula stage, often seen with direct cleavage. When considering embryo development in Group 2, a prolonged time interval between the 2- and 3-cell stages has been associated with a reduced likelihood of blastocyst formation [17] and lower implantation potential in humans [9]. Additionally, in cattle, direct cleavage (shown in many blastocysts in Group 3 in this study) has been associated with a higher likelihood of the embryo being chromosomally abnormal [29]. With this in mind, the resulting blastocysts in both Groups 2 and 3 are likely of lower quality than those in Group 1.

Using binomial linear regression, the finding that the time taken to reach the 2-cell stage is predictive of blastocyst formation (Figure 3) is significant. That is, because pronuclei cannot be visualised in dark, lipid-laden pig embryos and the overall cleavage rate is a poor predictor of blastocyst formation due to high polyspermy [47], the timing of first the cleavage is an alternative means of assessing developmental potential. It also correlates with prior studies indicating that early first cleavage is indicative of developmental potential [11,31,42]. Isom and colleagues performed an assessment of cleavage at 24 h, 30 h and 40 h post insemination, with the number of blastocysts per cleaved embryo on day 7 being 39.9 ± 0.8%, 24.6 ± 2.5% and 10.5 ± 1.3% respectively [11]. Using a similar approach, Booth and colleagues assessed embryos every two hours, showing that most embryos that reached the blastocyst stage (82%) had cleaved within 25 hpi [42]. However, this is the first study to use time-lapse technology and the first to empirically establish the extent to which the timing of the first cleavage is predictive of blastulation. We are conscious, however, of the speculative nature of these observations as inter-embryo developmental dynamics may be altered by many factors, such as parental genotype.

Of all embryos included in this study, 35% displayed chaotic cleavage, which proved to be lethal to embryo development. Chaos cleavage rates of around 15% in human embryos have been found. These embryos are not usually selected for cryopreservation and transfer and, when they are, show low blastulation rates [48]. In horses, chaotic cleavage can lead to blastulation and pregnancy at a lower level than normal cleavage [49,50,51]. It is likely that the high prevalence of chaotic cleavage seen in these embryos is associated with a high incidence of chromosomal abnormalities [28] in accordance with a recent human time-lapse study. Braga and colleagues reported a relationship between the timing of morphokinetic variables and aneuploidy rates, with the time taken to reach several key developmental time points being longer in aneuploid than euploid embryos [52]. Nonetheless, aneuploidy can also be associated with rapidly dividing embryos. Chaotic cleavage could also be related to polyspermy, the incidence and frequency of which is reported to be a major barrier to pig IVP success [8,53]. Interestingly, high rates of aneuploidy have been found in both in vivo and in vitro-derived pig embryos [54]. It is worth considering, therefore, that the embryos taking longer to reach key developmental milestones may be more likely to be aneuploid.

Significantly higher odds of an embryo reaching the blastocyst stage were seen in embryos displaying fragmentation up to 10% and between 11 and 25%. This is unsurprising, given that the majority of non-blastulating embryos exhibited more extensive fragmentation. It has been shown in several species that low levels of fragmentation (below 10%) are not associated with a decreased developmental potential but that, beyond 10%, the level of fragmentation is inversely correlated with resulting developmental potential and implantation rates [32,55].

Embryos exhibiting reverse cleavage had significantly increased odds of not reaching the blastocyst stage. These results are consistent with some limited studies in humans [56,57,58]. Conflict in the literature, however, exists with some clinical data showing that reverse cleavage cannot be used as a deselection criterion and that the presence of reverse cleavage is not associated with increased aneuploidy rates [59,60]. At the time of writing, the only comparable study in livestock indicated that reverse cleavage in cattle embryos reduced the incidence of hatching, increased blastocyst collapse and chromosomal aneuploidy [29]. Future studies should consider the ploidy status of embryos following different patterns of development and consider pregnancy and live birth outcomes. Moreover, it would be interesting to develop more complex regression models that could be tested for accuracy in predicting live birth outcomes and address potential boar and dam effects that could affect embryo morphokinetics.

## 5. Conclusions

Here, we highlight the benefits of morphokinetic analysis through time-lapse, with our most significant finding being that the time to reach two cells (the first cleavage) is the most accurate predictor of the likelihood of blastulation. This finding has an immediate application in that only those embryos that are first cleaving need to be retained, freeing up incubator space and providing a means through which the benefits and drawbacks of, e.g., novel media formulations could be investigated more rapidly.

## Figures and Tables

**Figure 1 animals-14-00783-f001:**
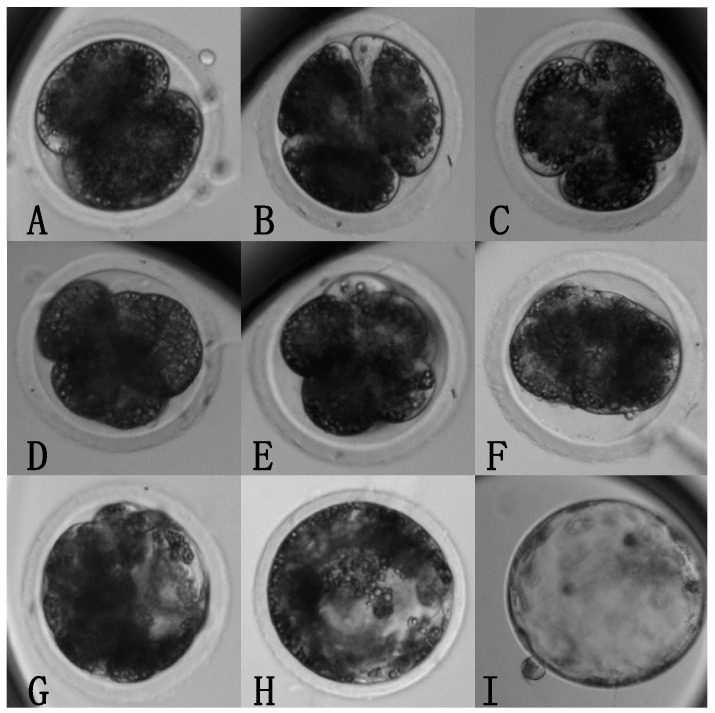
Key developmental events during early pig embryogenesis. Representative examples of nine key embryonic developmental events: (**A**): 2-cells; (**B**): 3-cells; (**C**): 4-cells; (**D**): 5+ cells; (**E**): compaction; (**F**): morula; (**G**): cavitation/early blastocyst; (**H**): expanding blastocyst; (**I**): hatching blastocyst. Images taken from a Geri^®^ Assess platform. The time taken to reach each of these events was recorded for each embryo. Magnification: 10×.

**Figure 2 animals-14-00783-f002:**
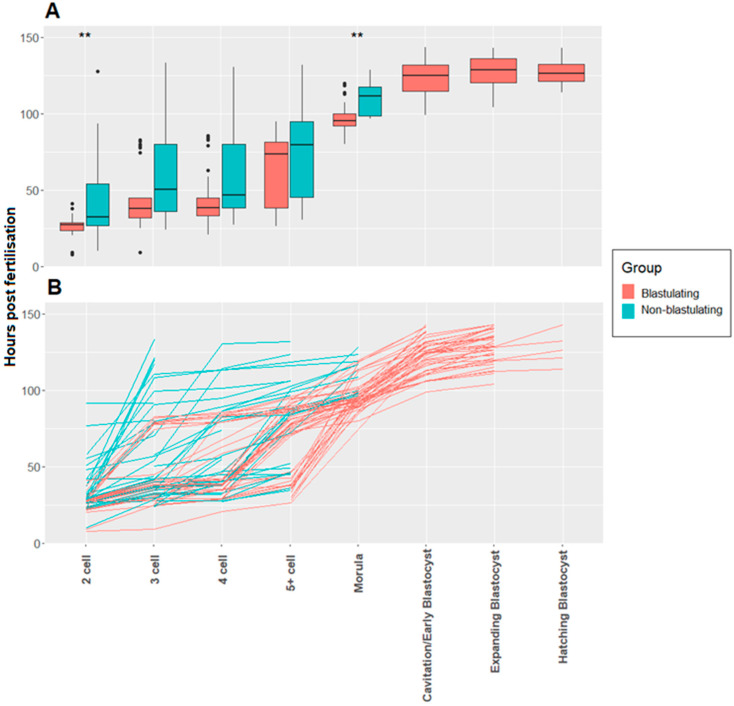
Comparison of the developmental kinetics of blastulating and non-blastulating in IVP pig embryos. ** denotes *p* < 0.01. (**A**): A boxplot showing the number of hours post-insemination (hpi) taken to reach each developmental stage for both blastulating (*n* = 47) and non-blastulating (*n* = 64) cleaved embryos. Analysis with a Mann U Whitney test with Bonferroni adjusted *p*-values showed a significant difference between the two groups for the time taken to the 2-cell (W = 370.5, *p* = 0.0015) and morula stage (W = 89, *p* = 0.0017). There was no significant difference found for the 3-cell (W = 460.5, *p* = 0.2), 4-cell (W = 381, *p* = 0.16) and 5+ cell stage (W = 453.5, *p* = 0.56). (**B**). Visualisation of the development of individual blastulating and non-blastulating IVP pig embryos. A line graph to show the time (hpi) taken to reach each developmental stage during early embryogenesis for both blastulating and non-blastulating (*n* = 111) cleaved IVP pig embryos.

**Figure 3 animals-14-00783-f003:**
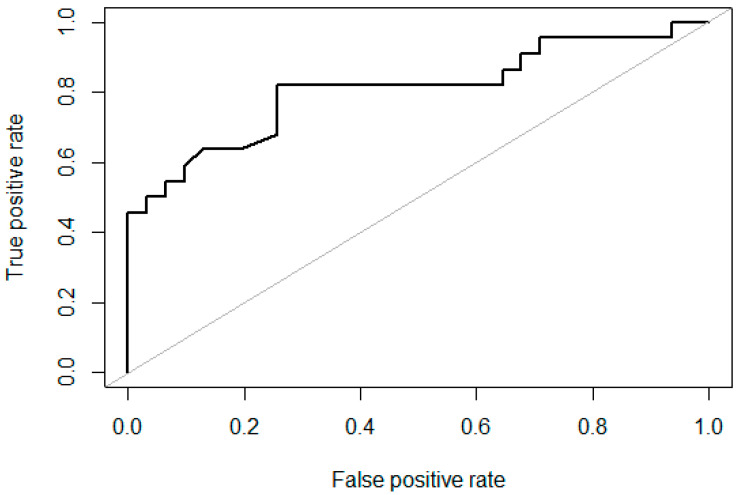
ROC curve and AUC score for the time taken to the 2-cell stage as a predictor of further development to the blastocyst stage. ROC = receiver operating characteristics curve. AUC = Area under the curve. AUC scores are interpreted as no discrimination = 0.5, acceptable discrimination = 0.7–0.8, excellent discrimination = 0.8–0.9 and outstanding discrimination = >0.9.

**Table 1 animals-14-00783-t001:** Intra-observer agreement for the tagging of morphological events during pig embryo development. Data from 51 embryos (after exclusion criteria were applied) were used to perform a two-way ICC to demonstrate intra-user agreement for the tagging of individual events during development. ICC values can be interpreted as follows: <0.5 poor agreement, 0.5–0.75 moderate agreement, 0.75–0.9 good agreement, 0.9+ excellent agreement. 95% confidence intervals (CI) are also given.

	Intra-Observer		
Developmental Event	ICC	*n*	95% CI
2-cell	0.998	38	0.996–0.999
3-cell	0.995	33	0.991–0.998
4-cell	0.999	30	0.997–0.999
5+ cell	0.998	32	0.995–0.999
Compaction	0.858	31	0.728–0.929
Morula	0.910	27	0.813–0.958
Early blastocyst	0.957	24	0.908–0.984
Expanding blastocyst	0.961	17	0.897–0.986
Hatching blastocyst	0.438	4	−0.759–0.952
Median ICC Mean ICC	0.961 0.902		

**Table 2 animals-14-00783-t002:** Use of the time taken to reach the 2-cell stage for the prediction of the likelihood of development to the blastocyst stage. Data is shown for embryos annotated at the 2-cell stage that either arrest or blastulate (*n* = 76).

Time Taken to 2-Cell Stage	Odds Ratio	*p*-Value
≤27 hpi	1.59	0.3
≤28 hpi	4.35	0.002
≤29 hpi	5.00	0.0008
≤30 hpi	10.00	0.00004
≤31 hpi	9.09	0.00008
≤32 hpi	9.90	0.0001

## Data Availability

Raw video files and associated data can be obtained by contacting katie.harvey@open.ac.uk.

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
