# Peer review of "Morphokinetic Profiling Suggests That Rapid First Cleavage Division Accurately Predicts the Chances of Blastulation in Pig In Vitro Produced Embryos"

_animals, 2024, doi:10.3390/ani14050783_

Round 1
Reviewer 1 Report
Comments and Suggestions for Authors
The authors describe the use of time-lapse incubator image technology to study embryo pig embryo developmental dynamics and correlations with early cleavage (2 cells) with subsequent morula and/or blastocyst formation.
Minor reviews:
Introduction
lines 47-75, the background information is redundant and repetitive, in one sentence please explain the benefit of an efficient IVP system in the pig.
Line 76-84, there is very little evidence that uninterrupted time lapse culture produces more competent embryos than a traditional well managed embryo culture incubator and embryo evaluation, please reword the sentence in the sense that uninterrupted culture may have theoretical advantages but there is no total consensus or evidence.
IVP
Line 134, how many boars were included in the study and was the boar effect included in the statistical model?, please clarify
Line 143, please clarify the volume of ivc medium per presumptive zygote, also specify if it was embryo group culture or individual embryo culture
Results
Line 191, the statement can be speculative as you are not addressing the male effect which plays a huge role in embryo developmental dynamics
Line 238, can you explain if morula compaction was associated with blastulation
Line 257, chaotic cleavage can lead to blastulation and pregnancy at a lower level than normal cleavage, in other species like horses. Please address the issue in the discussion
Disucussion
325 Please discussed the issue of rapidly dividing embryos and association with high rate of aneuploidy which contradicts your statement.
338, in horses recently a study indicates that reverse cleavage can lead to lower blastulation and pregnancy rates, please discuss accordingly
Explain why less than 29h to reach the two cell stage od ratio to blastulation is lower than 30-32 h. Your conclusion states that time to reach the two cell stage is predictive of blastulation.
Major concenrn:
Study design: multiple boars of different in vitro fertility should have been used and included in the statistical model.
It appears that only one boar perhaps of high in vitro fertility was used in the study. The effect of the male has profound consequences on resumption of meiosis, pronuclei formation, embryo development and dynamics and time of blastocyst formation. Therefore, the study is not representative of the population and the findings only apply to one boar.
Author Response
Reviewer 1.
The authors describe the use of time-lapse incubator image technology to study embryo pig embryo developmental dynamics and correlations with early cleavage (2 cells) with subsequent morula and/or blastocyst formation.
This is an accurate representation of our work.
Minor reviews:
Introduction
lines 47-75, the background information is redundant and repetitive, in one sentence please explain the benefit of an efficient IVP system in the pig.
We have reduced this significantly. It is however still a few sentences as we feel that this retains important detail. Lines 57-68.
Line 76-84, there is very little evidence that uninterrupted time lapse culture produces more competent embryos than a traditional well managed embryo culture incubator and embryo evaluation, please reword the sentence in the sense that uninterrupted culture may have theoretical advantages but there is no total consensus or evidence.
We have added a sentence to clarify this point. Lines 78-81.
IVP
Line 134, how many boars were included in the study and was the boar effect included in the statistical model?, please clarify
Six rounds of IVP were performed in this study, meaning that six batches of ovaries, and six semen samples were used – we have added this information to the paper (lines 125 and 141). The purpose of our work however was not to determine whether there was a boar effect on morphokinetic parameters; to do so would have required a very different experimental design. Whilst we agree that effects of morphokinetic parameters have been identified in comparisons of sperm origin when testicular and ejaculated sperm have been compared, we are not aware of studies that have looked at individual differences. It would be equally plausible to argue that variation between females could contribute in a similar manner. Statistical analysis of such effects would be very difficult, if not impossible, in this system. To this end, we have added some text to the manuscript to clarify that we do not address any potential boar effect.
Line 143, please clarify the volume of ivc medium per presumptive zygote, also specify if it was embryo group culture or individual embryo culture
The presumptive zygotes were (as noted in the methods, line 150 onwards) incubated in Geri® dishes that allow for the culture of up to 16 embryos in microwells. These are individual wells, but all embryos are cultured in the same 80µl droplet of culture media, that is overlaid with 4ml of mineral oil. None of the embryos come into contact with each other, but they are co-cultured in the same media. We have further clarified this in the manuscript.
Results
Line 191, the statement can be speculative as you are not addressing the male effect which plays a huge role in embryo developmental dynamics
We have now addressed this point in the discussion – lines 356-358.
Line 238, can you explain if morula compaction was associated with blastulation
As shown in Figure 2, the time taken to reach the morula stage (i.e., the time taken to compaction) was significantly different between embryos that blastulated and those that did not.
Line 257, chaotic cleavage can lead to blastulation and pregnancy at a lower level than normal cleavage, in other species like horses. Please address the issue in the discussion
We have added a sentence pertaining to this to the discussion. Lines 362-364.
Disucussion
325 Please discussed the issue of rapidly dividing embryos and association with high rate of aneuploidy which contradicts your statement.
We have done this. Lines 384-386.
338, in horses recently a study indicates that reverse cleavage can lead to lower blastulation and pregnancy rates, please discuss accordingly
We have done this. Lines 362-364.
Explain why less than 29h to reach the two cell stage od ratio to blastulation is lower than 30-32 h. Your conclusion states that time to reach the two cell stage is predictive of blastulation.
As only limited numbers of embryos reached the two-cell stage in less than 29 hours, ability to discriminate between fates is lower, i.e.: as the number of embryos in the less than 29-hour group is much smaller than the number of embryos in the less than 30-hour group, statistical confidence is much lower.
Major concenrn:
Study design: multiple boars of different in vitro fertility should have been used and included in the statistical model.
It appears that only one boar perhaps of high in vitro fertility was used in the study. The effect of the male has profound consequences on resumption of meiosis, pronuclei formation, embryo development and dynamics and time of blastocyst formation. Therefore, the study is not representative of the population and the findings only apply to one boar.
We have addressed this point in our response to the comment “Line 134, how many boars were included in the study and was the boar effect included in the statistical model?, please clarify” above.
Reviewer 2 Report
Comments and Suggestions for Authors
General comment
The research topic is innovative, as there have been no studies on the morphokinetics of pig embryos so far. Research related to embryo morphokinetics can be presented in a very attractive, transparent way (e.g. in sheeps, cats and cattle). However, this manuscript is very convoluted and difficult to understand. I suggest reediting the manuscript.
Detailed comments
Introduction
1. The introduction gives the impression of a conglomeration of various thoughts
2. In my opinion, there is too much information about ART in cattle, which is irrelevant to the topic.
3. I suggest focusing on;
- importance of ART in pigs (as in the paragraph 65-75)
- problems with ART in pigs (oocyte specificity, in vitro maturation of oocytes end embryo development) compared to other species)
- usability and advantages of the time-lapse systems ( this is well described in the manuscript)
- state of knowledge about morphokinetics in other species (cattle, sheep, cats, humans) and proving that there is a lack of this type of research in pigs.
Material and methods
How many ovaries (sows) were used?
Why did you incubate the oocytes with sperm for only 2 hours? Isn't that too short?
Results
The results are presented in a very difficult way, not typical for embryological research
Try to present in table the results in the classic embryological way:
- How many CoCs did You obtain?
- How many oocytes did You fertilize?
- Cleavage rate, morula rate, blastocyst rate , hatching rate n(%)
- percentage of individual morphological defects
Fig 1. I- If I'm seeing correctly, it's a hatching blastocyst, not an expanding one
Fig 2. - axis description „development time” is inaccurate, I suggest (hours post insemination/ferilization)
Discussion
276-282 - It is true that early time of the first division of the embryo is a good predictor of its further development. But note that very early division is not beneficial, which has been proven, for example, in humans and cats. Indicating the optimal time for the first division would be more appropriate. the indication that fastest is best is incomplete.
329 „…..In both human and pig embryos, it has been shown that low levels of fragmentation…” this has also been described in other species
Author Response
Reviewer 2.
Detailed comments
Introduction
- The introduction gives the impression of a conglomeration of various thoughts
We have shortened this as per the comments received from reviewer 1.
- In my opinion, there is too much information about ART in cattle, which is irrelevant to the topic.
We have shortened this as per comments from reviewer 1.
- I suggest focusing on;
- importance of ART in pigs (as in the paragraph 65-75)
Done.
- problems with ART in pigs (oocyte specificity, in vitro maturation of oocytes end embryo development) compared to other species)
Done.
- usability and advantages of the time-lapse systems ( this is well described in the manuscript)
Done.
- state of knowledge about morphokinetics in other species (cattle, sheep, cats, humans) and proving that there is a lack of this type of research in pigs.
This has been made clearer. Lines 99-103.
Material and methods
How many ovaries (sows) were used?
This is not data that we collected as part of this study, as ovaries were collected on the abattoir line. Any individual sow on that line may have contributed 0, 1 or 2 ovaries and these could not be related to individual sows.
Why did you incubate the oocytes with sperm for only 2 hours? Isn't that too short?
This is standard practice in pig IVP systems due to the increased incidence of polyspermic fertilisation in this species.
Results
The results are presented in a very difficult way, not typical for embryological research
Try to present in table the results in the classic embryological way:
- How many CoCs did You obtain?
- How many oocytes did You fertilize?
- Cleavage rate, morula rate, blastocyst rate , hatching rate n(%)
- percentage of individual morphological defects
This is standard practice in human embryology but is not for other species. It is our opinion that adding this would not help interpretation of the results presented here.
Fig 1. I- If I'm seeing correctly, it's a hatching blastocyst, not an expanding one
Yes, apologies, this was a mistake, and the figure legend has been changed.
Fig 2. - axis description „development time” is inaccurate, I suggest (hours post insemination/ferilization)
We have changed this.
Discussion
276-282 - It is true that early time of the first division of the embryo is a good predictor of its further development. But note that very early division is not beneficial, which has been proven, for example, in humans and cats. Indicating the optimal time for the first division would be more appropriate. the indication that fastest is best is incomplete.
It is not possible for us to accurately provide a range for the optimal time for the first division given our sample size (please see our reply to this comment above: “Explain why less than 29h to reach the two cell stage od ratio to blastulation is lower than 30-32 h. Your conclusion states that time to reach the two cell stage is predictive of blastulation”.
329 „…..In both human and pig embryos, it has been shown that low levels of fragmentation…”this has also been described in other species
We have made it clearer that this has been described in several species.
Reviewer 3 Report
Comments and Suggestions for Authors
Specific comments:
Lines 35 to 36: In the abstract you mention 2 hypotheses while in the introduction and the results section 3 are described instead.
Line 182: please, state clearly in the sentence that the 42.3% blastocyst rates you report are out of cleavage.
Line 183: Please change the sentence to the blastocyst rate per oocyte obtained was 17.5%.
Lines 191 to 205: Please explain figure 2A before 2B (or change the order of the figure). The three patterns of blastocyst development are not clearly explained, it is not easy to see it in the picture. When referring to number of hours, please add hpi (if that is what was measured), or the appropriate unit.
Figure 2B: Please make clear the three patterns explained in the text, because it is very difficult to understand it as it is.
Line 271: "of the earliest stages" is duplicated
Lines 286 to 290: In the discussion you name the different patters as group 1,2 and 3, which was not stated in the text before. Please include that classification in the results section.
References: 8 references are duplicated. 7=49, 9=57, 11=45, 13=31, 17=48, 24=61, 26=54, 27=51.
Author Response
Reviewer 3.
Specific comments:
Lines 35 to 36: In the abstract you mention 2 hypotheses while in the introduction and the results section 3 are described instead.
We have changed this.
Line 182: please, state clearly in the sentence that the 42.3% blastocyst rates you report are out of cleavage.
We have changed this – line 196.
Line 183: Please change the sentence to the blastocyst rate per oocyte obtained was 17.5%.
We have changed this – line 196.
Lines 191 to 205: Please explain figure 2A before 2B (or change the order of the figure). The three patterns of blastocyst development are not clearly explained, it is not easy to see it in the picture. When referring to number of hours, please add hpi (if that is what was measured), or the appropriate unit.
We have done this. Line 207 onwards.
Figure 2B: Please make clear the three patterns explained in the text, because it is very difficult to understand it as it is.
We have done this. Lines 210-213.
Line 271: "of the earliest stages" is duplicated
We have changed this.
Lines 286 to 290: In the discussion you name the different patters as group 1,2 and 3, which was not stated in the text before. Please include that classification in the results section.
This has been changed. Lines 210-213.
References: 8 references are duplicated. 7=49, 9=57, 11=45, 13=31, 17=48, 24=61, 26=54, 27=51.
The reference list has been updated and corrected.
Round 2
Reviewer 1 Report
Comments and Suggestions for Authors
Thank you for addressing the comments and suggestions
Author Response
Thanks for your contribution on the manuscript.
Reviewer 2 Report
Comments and Suggestions for Authors
Minor Revision
Please title the subsections more precisely
2.3 in vitro production (IVP) - production of what? -In vitro embryo production
If you use the abbreviation (IVP), consistently also use the abbreviation IVM in paragraph 2.2
2.4 in vitro culture in time lapse system – culture of what?
/„This is standard practice in human embryology but is not for other species. It is our opinion that adding this would not help interpretation of the results presented here” - /
Your answer is not polite and it is not true. For both humans and animals, it is standard to provide the number of patients (animals) or ovaries, the number of collected oocytes, the number of matured and ferilized oocytes etc. You may not have this data a, but your argument is wrong
In the methodology you write that you fertilized matured oocytes. How do you know they were matured?
Please just indicate
How did You select coocytes for IVF? Did You fertilize each oocytes after IVM?
How did you evaluate and prepare the oocytes for fertilization? Did You remove part of the cumulus
Author Response
Manuscript ID animals – 2863099
Morphokinetic profiling suggests that rapid first cleavage division accurately predicts the chances of blastulation in pig in vitro produced embryos
Authors: Lucy May Hillyear, Louisa J. Zak, Tom Beckitt, Griffin DK*, Simon Crawford Harvey, Katie Evelyn Harvey
Reviewer 2.
Minor Revision
Please title the subsections more precisely
2.3 in vitro production (IVP) - production of what? -In vitro embryo production
This has been changed.
If you use the abbreviation (IVP), consistently also use the abbreviation IVM in paragraph 2.2
This has been changed.
2.4 in vitro culture in time lapse system – culture of what?
This has been changed.
/„This is standard practice in human embryology but is not for other species. It is our opinion that adding this would not help interpretation of the results presented here” - /
Your answer is not polite and it is not true. For both humans and animals, it is standard to provide the number of patients (animals) or ovaries, the number of collected oocytes, the number of matured and ferilized oocytes etc. You may not have this data a, but your argument is wrong
We apologise that reviewer 2 thought our response to this comment was impolite – that was not our intention. We do not have these data, so we cannot add any further detail.
In the methodology you write that you fertilized matured oocytes. How do you know they were matured?
We fertilised all oocytes after IVM – we have made this clearer in the paper.
Please just indicate
How did You select oocytes for IVF? Did You fertilize each oocytes after IVM?
We state in the methods that “cumulus-oocyte complexes (COCs) were collected from follicles ranging from 3-8mm”, and that “COCs with three layers of cumulus cells and an even cytoplasm were selected”. We have made it clearer that oocyte fertilisation occurred after IVM.
How did you evaluate and prepare the oocytes for fertilization? Did You remove part of the cumulus
We have stated this in the paper: “IVM oocytes were washed twice in Porcine Gamete Medium (PGM) [34] and co-incubated with 100,000 sperm/ml for two hours. Oocytes were then moved to another fresh well of PGM for a further two hours to reduce the risk of polyspermy. Following this, presumptive zygotes were denuded by repeated aspiration using a P200 pipette.”